# The Finer They Get: Combining Fine-Tuned Models For Better Semantic Change Detection

**Wei Zhou**
University of Stuttgart
Bosch Center for Artificial Intelligence
wei.zhou@ims.stuttgart.de

**Nina Tahmasebi**
University of Gothenburg
nina.tahmasebi@gu.se

**Haim Dubossarsky**
Queen Mary University of London
Cambridge University
h.dubossarsky@qmul.ac.uk

## Abstract

In this work we investigate the hypothesis that enriching contextualized models using fine-tuning tasks can improve their capacity to detect lexical semantic change (LSC). We include tasks aimed to capture both low-level linguistic information like part-of-speech tagging, as well as higher-level (semantic) information.

Through a series of analyses we demonstrate that certain combinations of fine-tuning tasks, like sentiment, syntactic information, and logical inference, bring large improvements to standard LSC models that are based only on standard language modeling. We test on the binary classification and ranking tasks of SemEval-2020 Task 1 and evaluate using both permutation tests and under transfer-learning scenarios.

## 1 Introduction

The last few years have seen a growing interest in language change, specifically in lexical semantic change (LSC), from the NLP community. LSC is the linguistic phenomenon in which words' meaning may change over time (for example by adding senses, or broadening/narrowing in their meaning scope). Originally, the study of lexical semantic change aspired to understand the phenomenon from a linguistic perspective (Dubossarsky et al., 2016; Schlechtweg et al., 2017; Keidar et al., 2022). However, it was also motivated by the need for better handling of semantic change in other text-based research disciplines that work with historical texts (e.g., lexicographers, historians).

In more recent times, the understanding that general purpose NLP models also need to accommodate for the fluidity of word meaning has reached the greater NLP community, bringing with it the realization that LSC plays a vital role (Barbieri et al., 2022). It is particularly visible in the deterioration of model performance over time because the language on which models and other algorithms are (pre-)trained, starts to drift as time passes (Rosin and Radinsky, 2022; Alkhalifa et al., 2023). When deployed, these models process text from time periods they were not trained on, which hinder their performance.

This wide, multi-disciplinary interest in LSC, has driven the development of many computational models for language change (Kutuzov et al., 2018; Tahmasebi et al., 2021). In addition, much work has been devoted to supporting this progress by curating evaluation datasets that provide appropriate testing of these new models. Most of these datasets are in the form of SemEval tasks and contain high quality, humanly annotated lists of words. Each word has either changed its meaning between the considered time periods, or remained stable in meaning, and each list is accompanied by relevant historical corpora. This has become the de-facto evaluation standard in the field.

When reviewing the different models that are evaluated in SemEval, most of them use the same suit of methods that rely on standard distributional models of meaning. These models are either trained solely on historical text (e.g., SGNS or other static models), or use contextualized models pre-trained on a large "general purpose" text.[1] All of these models generate meaning representations in vector form for words from a historical corpus and compare them to vectors representing the modern meaning of the same word. Although the models differ (e.g., in terms of data or training objectives), they all share the same basic trait - they rely on meaning representation based on neighboring words without additional linguistic information.

---

[1]Some contextualized models are also fine-tuned on historical corpora.

This state of affairs raises a question: do models that are based solely on collocation statistics and trained exclusively on a masked-word prediction task suffice for our purpose? That is, are classic distributional models able to capture the full repertoire of word meaning, and then access it when analyzing meaning change? In this work, we investigate several concerns that we think a-priori suggest that additional linguistic information is beneficial for unsupervised lexical semantic change detection.

One direct path forward is fine-tuning pretrained models on additional tasks or domains like Question Answering and Sentiment Analysis. It has been demonstrated that fine-tuning of models helps even when the fine-tuned tasks are different than the target task (for example, fine-tuning on textual summarization and testing on Question-Answering) in a sort of a zero/few-shot transfer learning (Peters et al., 2019; Merchant et al., 2020; Khashabi et al., 2020). Therefore, it is reasonable to assume that enriching a contextualized model with additional fine-tuning tasks would lead to improved performance also for LSC detection.

In this paper, we test this hypothesis with one of the top-performing models of LSC detection in English and explore the potential to improve it by enriching it through a set of fine-tuning tasks. We provide our code here.[2]

## 2 Related literature

### 2.1 Models of LSC and their evaluation

In the past years, we have seen an increasing amount of models for unsupervised detection of lexical semantic change, almost exclusively focused on distributional semantic models. SemEval-2020 Task 1 was the first attempt at a large-scale evaluation and comparison of methods on four different languages. Two main classes of methods were evaluated by the participating teams. The first was based on type embeddings, either those that require alignment between independently trained models, (e.g., SGNS with Orthogonal Procrustes alignment (Arefyev and Zhikov, 2020)) or static embeddings without the requirement of alignment (Zamora-Reina and Bravo-Marquez, 2020). The second class was based on contextualized embeddings and combined with e.g., a clustering algorithm to derive sense information (e.g., XLM-R with K-means clustering

(Gyllensten et al., 2020)) or other means of comparing vectors in each time period with each other (Kutuzov and Giulianelli, 2020). The models were evaluated on two tasks, binary classification and ranking.

For the four SemEval datasets, the trend was that type-based models outperformed contextualized (token) models. It was also clear that good performance on binary classification does not necessarily indicate good performance on the ranking task. Since SemEval-2020, more evaluation tasks have been curated for Russian (Kutuzov et al., 2021), Italian (Basile et al., 2020), Norwegian (Kutuzov et al., 2022a) and Spanish (Zamora-Reina et al., 2022), where we see stronger indications that contextualized models perform better than type-based ones.

### 2.2 Contextualized models training

Training contextualized models requires a massive amount of textual data, prolonged training time, and considerable computational power. All these have made the training of new models a complicated procedure available only to selected research labs over the world, as oftentimes researchers lack the necessary resources to train their own models (our interest in historical language poses a particular challenge in this regard, as historical texts are usually much smaller in size).

To mitigate these requirements, speed up the training process, and increase the usability of these models, *fine-tuning* was developed. Using fine-tuning, models that were already trained (now called pre-trained models) are continued to be trained, albeit on much smaller data and sometimes on a different task (pre-trained models are usually trained on standard masked-word and next-sentence prediction tasks).

This two-step training setup was found to greatly improve the state-of-the-art performance in many tasks and today, the use of fine-tuning in contextualized models has become the dominant paradigm in NLP (Howard and Ruder, 2018; Devlin et al., 2019; Merchant et al., 2020). Importantly, it was found that fine-tuning, henceforth FT, can *transfer* to other tasks and languages and thus improve performance on tasks and datasets it was not trained on (Peters et al., 2019; Khashabi et al., 2020), presumably because of shared information that is required to process these different tasks. In this paper, we aim to leverage the trans-

---

[2]https://github.com/ChangeIsKey/LSC-AGG

fer capabilities and test whether FT a contextualized model on a range of NLP tasks improves its performance to detect LSC.

## 3 Method

Our aim is to test whether a state-of-the-art method for detecting lexical semantic change, based on pre-trained contextual embeddings, can be improved by adding fine-tuned layers. Therefore, we start with a BERT model and detect semantic change following Kutuzov and Giulianelli (2020). We chose their model as it was the best-performing system in English in the post-evaluation stage of SemEval-2020 Task 1. Next, we add information from fine-tuning for tasks that go beyond a masked language model objective. We include tasks aimed to capture both low-level linguistic information like part-of-speech tagging, as well as higher-level (semantic) information such as sentiment analysis, linguistic inference, and machine reading comprehension.

### 3.1 LSC method

The task of detecting lexical semantic change can be described as the following: given two corpora $C_1$ and $C_2$ from time periods $T_1$ and $T_2$, as well as a set of target words, detect which words have changed between $T_1$ and $T_2$ as evidenced in $C_1$ and $C_2$. This is a special case of the general LSC problem which includes arbitrarily many time periods $T_1, \ldots, T_N$.

Following Kutuzov and Giulianelli (2020), we use a pre-trained BERT base model to generate the contextualized embeddings of each occurrence of the target words in $C_1$ in $C_2$, resulting in two corresponding embedding matrices $U_w^{t1}$ and $U_w^{t2}$. Given these embedding matrices, we calculate the change scores of each target word in one of two ways: inverted cosine similarity over word prototypes (PRT); and average pairwise cosine distance between token embeddings (APD).

$$PRT(U_w^{t1}, U_w^{t2}) = \frac{1}{d(\frac{\sum_{x_i \in U_w^{t1}} x_i}{N_w^{t1}}, \frac{\sum_{x_j \in U_w^{t2}} x_j}{N_w^{t2}})} \quad (1)$$

$$APD(U_w^{t1}, U_w^{t2}) = \frac{1}{N_w^{t1} N_w^{t2}} \sum_{x_i \in U_w^{t1}, x_j \in U_w^{t2}} d(x_i, x_j) \quad (2)$$

$N_w^{t1}, N_w^{t2}$ stands for the number of occurrences of $w$ in $T1$ and $T2$. $d$ is the cosine distance. For both methods, higher values suggest a larger semantic change.

### 3.2 Fine-tuning

The main contribution of this paper is the injection of richer meanings into contextualized embeddings using fine-tuning. Our fine-tuned models are derived mostly from adapters (Pfeiffer et al., 2020; Poth et al., 2021), which are trained layers that can be integrated directly into transformer-based models (the most popular type of contextualized models) in order to perform different tasks. Using adapters enabled us to speed up our experiments as they are readily available[3] and can be seamlessly integrated into the tested models.

In addition to using adapters, we also fine-tune two models locally on sentiment classification and part-of-speech tagging in order to compare the performance of fine-tuned models with adapter-based models. For sentiment classification, we use the sst2 dataset (Pang and Lee, 2004) while for part-of-speech tagging we use CoNLL2003 (Tjong Kim Sang and De Meulder, 2003). Since there is no test set with gold labels for sst2, we randomly sample 30% of the data from the validation set as a test set. The accuracy of the fine-tuned models on the test set is 0.908 for sentiment analysis and 0.931 for part-of-speech tagging. We use the BERT-base-uncased model for all our experiments, both with adaptors and local FT. Table 1 details the FT tasks we used.

| Task/Model | Type |
| --- | --- |
| natural language inference (nli) | pf |
| machine reading comprehension (reading compre) | pf |
| sentiment (sst2) | pf |
| sentiment (sst2-pfeiffer) | pfeiffer |
| sentiment (sst2-hously) | hously |
| semantic textual similarity | pf |
| linguistic acceptability | pf |
| grammatical error correction (error detect) | pf |
| semantic tagging | pf |
| named entity recognition (ner) | pf |
| part-of-speech tagging (pos) | pf |
| phrase chunking | pf |
| sentiment (sst2-fine-tune) | fine-tuned |
| part-of-speech tagging (pos-fine-tune) | fine-tuned |

Table 1: Fine-tuned models & tasks for adapters (upper) and locally trained FT (lower). Type refers to adaptors trained by Poth et al. (2021)(pf), Peiffer and Hously. Task abbreviations in parentheses.

---

[3] https://adapterhub.ml/

## 4 Evaluation

For our experiments, we use a standard evaluation dataset for LSC.

### 4.1 Evaluation data

We use the English dataset of SemEval-2020 Task 1 (Schlechtweg et al., 2020) for unsupervised lexical semantic change detection. The task was the first of its kind to provide manually annotated gold data for the purpose of fair and comparable evaluation of methods for LSC. The task consists of two sub-tasks aimed to measure change between two time-specific corpora $C_1$ and $C_2$:

**Binary Classification**: for a set of target words, decide which words lost or gained sense(s) between $C_1$ and $C_2$, and which did not.

**Ranking**: rank a set of target words according to their degree of LSC between $C_1$ and $C_2$.

These tasks are related but complementary. The ranking task measures the degree of change and takes into consideration lost or gained senses, but also includes changes in existing senses (e.g., by means of broadening or narrowing) which the binary classification task does not consider.

The English dataset of SemEval consists of 37 target words derived from the Clean Corpus of Historical American English (CCOHA) (Davies, 2012; Alatrash et al., 2020). The two 50-year periods are $C_1 = 1810-1860$ and $C_2 = 1960-2010$ from which each target word has a set of 100 randomly sampled sentences. These sentences have been compared by human annotators and ranked on a scale for lexical semantic change. Based on the outcome of the roughly 29,000 human judgments, words are classified as changing or stable and assigned a change degree. The process is described in detail by Schlechtweg et al. (2021)

### 4.2 Evaluation Metrics

We use two evaluation metrics.

**Spearman correlation** is used to compute the rank correlation between model predictions and gold labels in the ranking task.

**AUC & ROC** are used to evaluate the impact of different thresholds on the model performance in the binary classification task.

### 4.3 Validation through permutation

The work presented in this paper suggests that certain combinations of FT tasks improve LSC detection for both ranking and classification tasks. Importantly, these combinations are chosen based on improved performance in the SemEval tasks. Ideally, we would consider this as the training set and then test the chosen combinations on a held-out dataset to examine if similar gains are acquired. However, such a test set is lacking, and cannot be constructed via standard train-test splits from the 37 words, from which only 16 have changed.[4] We propose permutation tests to mitigate this shortcoming, and to enable us to draw reliable conclusions from our study despite this limitation.

For both tasks, we evaluate the probability that the best combinations we report were found by chance. We conduct a permutation analysis and generate artificial FT task scores that are based on the distributions of the existing FT values from the 14 FT models (Table 1). We then compute the relevant evaluation metric (Spearman or AUC) for each artificial FT, and repeat the process 100,000 times, creating a distribution of ranking or classification performance scores. We then compute the proportion of times that the artificial random FT combinations performed better than our best combinations, in the form of a p-value for our chosen combinations. Ultimately, this allows us to test the statistical significance of our results, and evaluate how likely it is that our best combinations were found by chance.

## 5 Experiments

Two models are used as baselines, relative to which we test if adding FT (either adaptors or local FT) improves performance. One of the baseline models was used as the basis on top of which the different FT combinations were tested.

We choose the best combination of FT tasks by analyzing the results of the *ranking task* and then test it on the *binary classification task* using a modified decision criterion (i.e., ROC analysis). Because the two tasks are different, this allows us to use the latter as an ad-hoc evaluation test.

---

[4]Under these conditions it was also not feasible to conduct a systematic regression analysis, which would have lacked the statistical rigor to reach reliable results.

## 5.1 Baselines

We make use of two rather different baseline models that are used together with the fine-tuning. Kutuzov and Giulianelli (2020) whose model scored the highest in the English part of SemEval (henceforth BERT), and HistBERT (Qiu and Xu, 2022) which provides a contextualized model pre-trained on historical English.[5] Together, they provide complementing baselines to test our research hypothesis. All FT combinations where made on top of the BERT baseline.

We also compute p-values from the permutation tests for the two tasks (see Section 4.3), and for each method (APD and PRT).

## 5.2 Ranking task

Given a fine-tuning task $FT_i$, we obtain two embedding matrices $U_w^{t1}$ and $U_w^{t2}$ for each target word. We use these embedding matrices to calculate the semantic change score of a target word by means of APD and PRT. Once we have the change scores for all target words, we produce a ranking of the words. We then measure the performance as Spearman correlation of the change score ranks compared to the gold ranking. This is illustrated in the following formula, where $P_{ind}$ stands for the performance of the individual task. $Score$ is the scoring function (AUC or Spearman correlation).

$$P_{ind} = Score(FT_i, gold) \qquad (3)$$

There are in total $FT_1, \ldots, FT_{14}$ fine-tuning tasks. In addition to their individual performance, we are interested if they add complementary information, and therefore want to measure their combined performance. We thus enumerate all the possible combinations of the tasks. For each combination, we average the change scores produced by each participating $FT_i$. For instance, we can combine change scores derived from Natural Language Inference and Named Entity Recognition by averaging the scores for each target. There are in total $FT_{14}^1 + FT_{14}^2 + \ldots + FT_{14}^{13} + FT_{14}^{14}$ combinations. We measured the performance of each combination by means of its Spearman correlation. This is illustrated in the following formula, where $c$ is the combination of different tasks.

$$P_c = Score\left(\frac{1}{|c|}\sum_{i \in c}(FT_i), gold\right) \qquad (4)$$

| Method | Rank-PRT | Rank-APD |
|---|---|---|
| ner | 0.218 | 0.285 |
| nli | **0.427**** | 0.634 |
| pos | 0.205 | 0.205 |
| error detect | 0.352 | 0.593 |
| linguistic acceptability | 0.364 | 0.622 |
| phrase chunking | 0.076 | 0.185 |
| pos-fine-tune | 0.277 | 0.087 |
| reading compre | 0.416 | 0.636 |
| semantic tagging | 0.265 | 0.255 |
| sst2 | 0.422 | 0.608 |
| sst2-hously | **0.435**** | 0.627 |
| sst2-fine-tune | 0.123 | 0.210 |
| sst2-pfeiffer | 0.391 | 0.459 |
| textual similarity | 0.378 | 0.694 |
| BERT | 0.423 | **0.706** |
| HistBERT-ave | 0.264 | 0.441 |

Table 2: Spearman correlations for different FTs on the ranking tasks. Best FTs in bold. Statistical significance marks *, **, ***: for p-values<.05, .01, .001, respectively.

## 5.3 Binary Classification task

For the binary classification task, we calculate the AUC score of each $FT_i$. One advantage of the AUC score over accuracy is that we do not need to define the threshold to determine a word changes or not, given the change scores derived from PRT and APD are continuous values

We carry out two experiments here: 1) testing the best models found in the ranking task on the binary classification task, and 2) examining the performance of individual $FT_i$ as well as combined models. Our motivation for the first experiment is that we want to evaluate our best models from a new perspective given that the ranking and classification tasks feature different task profiles. In the second experiment, we focus on the combination effect, and take the more challenging case, examining the FTs with the highest as individual FT tasks.

## 6 Results

### 6.1 Ranking task results

We begin by reporting the results of individual $FT_i$. The results are presented in Table 2. For PRT, the range of correlation between (the ranking produced by) each $FT_i$ and the gold rank is

---

[5]There are four versions: HistBERT-prototype, HistBERT-5, HistBERT-10, and HistBERT-full. They differ in the size and time period of the training data. In this study, we report the averaged scores of the four models.

| Method | Combination(s) | Correlation |
|---|---|---|
| APD best 5 | BERT, nli, textual similarity, error detect , pos-fine-tune | 0.723*** |
| | BERT, sst2, error detect, pos-fine-tune | 0.722*** |
| | sst2, textual similarity, error detect | 0.722*** |
| | BERT, sst2, textual similarity, error detect | 0.721*** |
| | BERT, textual similarity, error detect | 0.721*** |
| APD baseline | BERT | 0.706 |
| | BERT, random scores | 0.462 |
| | HistBERT (averaged) | 0.441 |
| PRT best 5 | nli, pos-fine-tune, sst2-fine-tune | 0.531** |
| | nli, sst2-pfeiffer, pos-fine-tune, ner, sst2-fine-tune | 0.515* |
| | BERT, nli, sst2-pfeiffer, ner, sst2-fine-tune | 0.503* |
| | nli, sst2-pfeiffer, pos-fine-tune, sst2-fine-tune | 0.503* |
| | nli, reading compre, sst2-pfeiffer, histBERT-10, pos-fine-tune, sst2-fine-tune | 0.502* |
| PRT baseline | BERT | 0.423 |
| | BERT, random scores | 0.336 |
| | HistBERT (averaged) | 0.264 |

Table 3: Ranking results for 5 best FT combinations, APD and PRT. p-values as reported in Table 2.

0.427 – 0.076. We find that most fine-tuned models do not beat the BERT baseline, with only 2 exceptions (nli and sst2-hously). For APD, individual $FT_i$s range from 0.694 to 0.086, with one FT having a negative correlation of 0.211. Here, the maximum baseline (BERT) is marginally higher than any individual FT with a correlation value of 0.706. Overall, the results from Table 2 show that most individual FTs do not improve task performance further for both PRT and APD.

We now turn to combining different FT for the ranking task, shown in Table 3. For PRT, the five best models (0.531 – 0.502) all rank higher than the baseline models (0.423 – 0.264). For APD, the five best models (0.723 – 0.721) also rank higher than the baseline models (0.706 – 0.441).

We note that although the performance gains are statistically significant for both PRT and APD, they are much more prominent for PRT. We also note that not every combination leads to an improvement. Some tasks (or task combinations) can yield lower performance. For instance, combing part-of-speech and sentiment in APD actually deteriorates task performance. More details can be found in the appendix.

The random permutations, where we average BERT with scores sampled from the overall score distribution, over 100,000 runs, corroborate our findings. For PRT we get a mean correlation of 0.336 (s.d. of 0.107), and less than 1 per 100 randomly sampled scores perform better than the best PRT (0.531) (p-value<0.01). For APD, the corresponding values are 0.462 (s.d. of 0.147), and less

than 2 per 1000 randomly sampled scores perform better (p-value<0.01).

## 6.2 Classification task results

| Method | AUC-PRT | AUC-APD |
|---|---|---|
| ner | **0.688*** | 0.604 |
| nli | 0.646 | 0.714 |
| pos | 0.634 | 0.536 |
| error detect | 0.634 | 0.670 |
| linguistic acceptability | 0.628 | 0.676 |
| phrase chunking | 0.622 | 0.622 |
| pos-fine-tune | 0.634 | 0.643 |
| reading compre | 0.670 | 0.696 |
| semantic tagging | 0.649 | 0.631 |
| sst2 | 0.658 | 0.664 |
| sst2-hously | **0.682*** | 0.685 |
| sst2-fine-tune | **0.673*** | 0.417 |
| sst2-pfeiffer | 0.631 | 0.613 |
| textual similarity | 0.661 | **0.741** |
| BERT | 0.673 | 0.717 |
| HistBERT-ave | 0.657 | 0.659 |

Table 4: AUC scores for different FTs on the classification task. Best FTs in bold. p-values as reported in Table 2.

We begin by reporting the results of individual $FT_i$, shown in Table 4. We report the AUC scores to avoid threshold selection. We observe more variance in APD than in PRT in terms of model performance. For PRT, we observe that named entity recognition provides the highest AUC score (0.688) while phrase chunking generates the lowest performance (0.622). As for APD, textual similarity was the only FT to surpass the baseline and achieve statistical significance with AUC score of

| Method | Combination(s) | AUC |
|--------|----------------|-----|
| APD best 3 | nli, textual similarity | 0.741* |
| | textual similarity, nli, BERT | 0.738* |
| | textual similarity, BERT | 0.735* |
| APD ind best | textual similarity | 0.741* |
| PRT best 3 | ner sst2-hously, sst2-fine-tune *(p-value = 0.054)* | 0.732 |
| | ner, sst2-fine-tune, reading compre *(p-value = 0.090)* | 0.720 |
| | ner, sst2-hously, sst2-fine-tune, reading compre *(p-value = 0.161)* | 0.714 |
| PRT ind best | ner *(p-value = 0.161)* | 0.688 |

Table 5: Results of combining different FTs on the classification task. We present the best individual task performance (APD/PRT individual best) for comparisons. * means statistically significant improvement over the BERT baseline.

0.741. Similar to what we found for the ranking task, it seems that most individual FTs do not improve task performance neither for PRT or APD.

To test the performance of combinations, we take the top 5 best-performing models for PRT and APD separately. We then enumerate all possible combinations from them and report the results in Table 5. We find that combining different FTs improves task performance in PRT but not in APD. In PRT, we see a five percentage of AUC increase when we combine Name Entity Recognition, sst2-hously and sst2-fine-tune (from 0.688 to 0.732). In APD, we do not observe performance gains over the individual FTs.

### 6.3 Transferability scenario

As the ranking and classification tasks are curated differently (see Section 4.1), we see this as an opportunity to use the latter as an ad-hoc evaluation set by testing the transferability between the two tasks. We ask: are the best combinations that we found for ranking also useful (i.e., can be transferred) for the purpose of classification? We apply the three best combinations found in Section 6.1 to the classification task, plot their ROC curves and calculate their AUC scores. The results are shown in Figure 1. We find that the best models found in the ranking task outperform BERT baseline in the classification task. In PRT, the best model achieves an AUC score of 0.774 compared with the base model (0.673). In APD, though less obvious, the best model still performs better than the baseline (0.744 v.s. 0.717). While the ranking and classification tasks are designed for different purposes, this experiment suggests that the analysis of ranking results can guide the choice, and thus transfer, of models for the classification task.

## 7 Discussion and Conclusion

In this paper, we investigate our hypothesis that adding linguistic information to pre-trained language models by means of fine-tuning can lead to improved performance on unsupervised Lexical Semantic Change detection. We chose two classic LSC tasks, ranking and binary classification, from SemEval-2020 Task 1 (Schlechtweg et al., 2020). To simplify and speed up the process of fine-tuning we used adaptors (Pfeiffer et al., 2020), which are pre-trained fine-tuning modules that can augment existing contextualized models and are readily and freely available for English.

First, we tried single FT tasks, which showed little or no improvement over the BERT baseline. Then we combined several FT tasks together by means of simple averaging and found considerable improvements.

Adding linguistic information, like part-of-speech, to a standard (masked) language model can offer additional information that improves the ability of the models to detect lexical semantic change. However, some kinds of information are adjacent to semantic change and therefore make the models capture change, but not necessarily semantic change. In future work, we will conduct in-depth analysis of the worst models, to see what information they capture instead and why this information seems to hurt the performance of the LSC model.

Our work is not the first to introduce linguistically augmented contextualized models for the task of LSC. Giulianelli et al. (2022) used an ensemble method to inject linguistic information, and reported performance gains in LSC tasks. However, they focus on "low-level" morpho-syntactic information. Our approach, in addi-

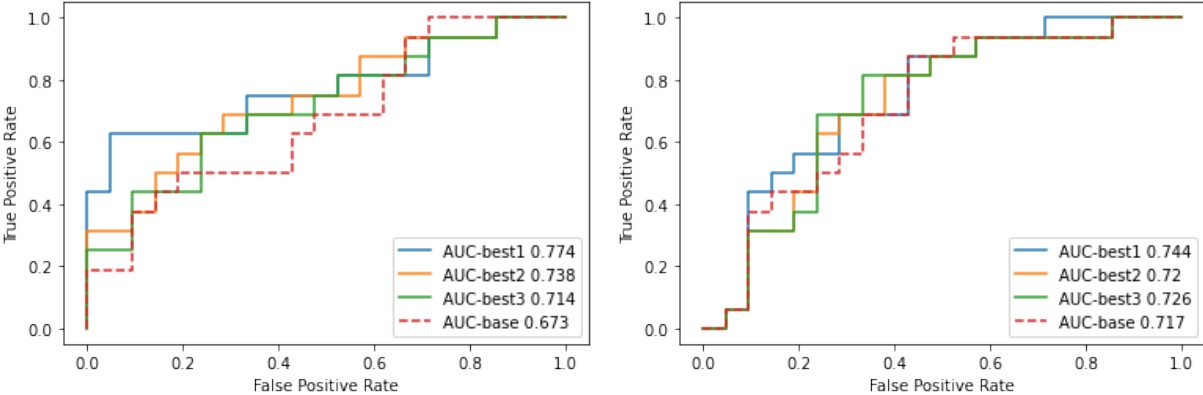

Figure 1: ROC curves of the best 3 models in the ranking task as they perform on the classification task. Left: PRT, right: APD. See Table 3 for combinations details.

tion to using a completely different ML method for linguistic augmentation, spans both ends of the linguistic-informative spectrum, ranging from part-of-speech, to sentiment to logical inference.

One deficiency of our results is that they are based on a small evaluation dataset, which means that the improvements we report could be attributed to chance (or model over-fitting). To mitigate this concern, and add scientific rigour to our analyses, all results were tested in permutation tests and are reported with their p-values. Evaluation is also done with comparison to two strong baseline models, each of which provides different perspective to test our research hypothesis. Outperforming the Kutuzov and Giulianelli (2020) model suggests that enriching models with additional linguistic information is highly beneficial, and outperforming HistBERT supports the idea that this information cannot be gained even when a model is exclusively (pre-) trained on historical text. Overall our results clearly suggest that LSC models can be improved dramatically with relatively simple steps of fine-tuning on a range of standard linguistic tasks.

We note that there are differences between the best performing FTs for the PRT and APD methods. Although certain FTs are shared (e.g., NLI and POS), others appear more systematically in PRT or APD. We interpret these inconsistencies as stemming from differences between the PRT and APD methods themselves, and do not view them as negative. Instead, each method enriches the baseline with different types of information and hence allows the model to capture slightly different aspects of LSC. Combined with the FTs, the final results can be quite different. This comple-

menting view of the two LSC methods is supported by the results of Kutuzov et al. (2022b), who recently reported that joining PRT and APD improves LSC detection results. The most probable explanation for this is that the two LSC methods are sensitive to different aspects of change.

From a theoretical point of view, our conclusions are inline with how linguists describe the phenomenon of LSC. Linguistic theory distinguishes between different types of LSC, and emphasizes that changes are never "general" but pertinent to certain aspects of meaning. Therefore, computationally analyzing words' meaning change using a "single ruler" as is done by current state-of-the-art LSC models, may be insufficient to describe the richness and diversity of change. We believe our findings provide an inroad for extending the capacity of LSC models and encouraging future research in this direction.

This is but the first step in exploring the potential of using FT to enrich and improve contextualized models of LSC. In our future work we will corroborate these findings more rigorously: extending these to other languages, testing the generalizability of the chosen FT tasks across LSC models, and test our approach in the discovery of new cases of words that change their meaning to go beyond a small set of examples.

## Acknowledgement

This work was supported in part by the Riksbankens Jubileumsfond (under reference number M21-0021, *Change is Key!* program), and in part by the Swedish Research Council (2019–2022; contract 2018-01184, *Towards Computational Lexical Semantic Change Detection* project).

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

# A Appendix

| Method | Combination(s) | Correlation |
|---|---|---|
| APD worst 3 | pos-fine-tune, sst2-fine-tune | -0.214 |
| | sst2-fine-tune | -0.210 |
| | pos-fine-tune | 0.087 |
| APD baseline | BERT | 0.706 |
| | BERT, random scores | 0.462 |
| | HistBERT (averaged) | 0.441 |
| PRT worst 3 | sst2-fine-tune, histBERT (full) | 0.012 |
| | pos-fine-tune, sst2-fine-tune, histBERT (full) | 0.014 |
| | pos-fine-tune, histBERT (full) | 0.038 |
| PRT baseline | BERT | 0.423 |
| | BERT, random scores | 0.336 |
| | HistBERT (averaged) | 0.264 |

Table 6: Ranking results for 3 worst FT combinations