# OpenReview forum: "The Finer They Get: Combining Fine-Tuned Models For Better Semantic Change Detection"
_NoDaLiDa/2023/Conference — NoDaLiDa 2023_

### Official Review · Reviewer_BzPR · 2023-03-07
**Interesting improvements on lexical semantic change detection with pre-trained language models**

**Rating:** 8
**Confidence:** 5

**Review:**

The paper proposes to "upgrade" the existing semantic change detection (SCD) methods based on pre-trained language models. The proposed improvement is basically adding combinations of adapters fine-tuned for various NLP tasks to the models. The authors show that this robustly improves SCD for English.

I generally like the paper. The proposed method is simple, linguistically motivated and efficient. One question I have is why the authors use the PRT and APD algorithms only separately? [Kutuzov et al 2022] have shown that averaging PRT/APD yields generally better results (check https://nejlt.ep.liu.se/article/view/3478). would be interesting to see whether your finding with adapters hold for the PRT/APD ensemble as well.

The main motivation behind the paper is "to enrich the existing SCD models with linguistic information". In this respect, it is strange that this paper is not mentioned: https://aclanthology.org/2022.lchange-1.6/
They move along the same lines (enriching language models with linguistic data for SCD) but in a slightly different way (using grammatical profiles changing over time). Would be interesting to compare to your approach - or at least mention the paper.

Some minor comments:

1) Talking about both SGNS and masked language modeling on line 91-101 is a bit weird. I guess word2vec training objective can in principle be described as MLM, but it would be a very non-standard definition: usually MLM is associated with  BERT-like encoder models, not word2vec-like type word embeddings. Better rephrase.

2) l.159 - Where does "100 vectors" come from? Why 100? The SemEval'20 English word usage graphs (WUGs) were indeed created from 100 examples per word per time period, but the LSC systems in the shared task were not limited to these 100 samples (in fact did not even have access to them). They operated on all the occurrences of the target words in the corresponding historic corpora (hundreds and thousands of them).

3) In the equation 2 (l.263), d is actually cosine _distance_, not cosine _similarity_.

4) l.696: simply --> simplify

**Paper Type:**

Long paper

---

### Official Review · Reviewer_rUys · 2023-03-12
**Solid research on fine-tuning and language change detection, presentation can be improved**

**Rating:** 7
**Confidence:** 4

**Review:**

The paper provides a range of experiments that investigate whether fine-tuning a BERT model on specific tasks can help improve detection of semantic change. The paper looks both at impact of individual tasks and by combining the outcome of multiple tasks by averaging over the detected change of each model. The results show that models fine-tuned on one task mostly don't improve on the baseline model, but results of combined models do. The paper also shows that results differ depending on whether one evaluates on ranking output according to their degree of change or whether the task is treated as a binary classification task (change or not).

The presented setup is solid and results are interesting, with as an only note that the evaluation data is very small as discussed in the paper. The paper is well-written in various parts, but there are also parts that I found rather confusing. Since I think I managed to figure everything out, I recommend the paper to be accepted. I recommend a thorough revision of some of the sections to make the message clearer, which may also help increase the impact of the paper. In particular, how does the outcome relate to the hypothesis exactly and what does it mean that averaging over certain tasks does help improve where individual fine-tuning usually does not?

More detailed feedback:

- When presenting the baselines, it would be clearer if it were made explicit that one of the baselines is the model that you are also fine-tuning.
- Also in Section 5.1 `the only contextualised model on historical English', MacBerth [1] is also trained on historical English. (In general, it is good practice to hedge statements on something being the only or the first or similar claims with `to (the best of) our knowledge'.

I found the Results Section relatively difficult to follow. I recommend a rewriting this section starting from the main point that you would like to get across and keeping a reader who was not involved in the experiments in mind. More specifically:
- it would be helpful to introduce the two experiments in Section 5 in the same order as the results are presented, this avoids confusion by the order being flipped.
- L513: `Interestingly,': why is this interesting? The fact that these couple of tasks do yield better results or that the others don't? If so, why is this particularly interesting? As a reader, I get the feeling I am missing something.
- in general in that paragraph: a perfect system would correlate perfectly, right? By presenting the higher correlations first and the formulation of how the two results relate to the baseline, I started to second guess this. I recommend simply describing the observations (most fine-tuned models do not beat the baseline, with 2 exceptions...) and then add a sentence that explicitly states how this relates to the hypothesis.
- assuming an additional page is granted upon acceptance: maybe split the results so that it is not necessary to go back and forth between tables. As a reader, you are already looking at the results of the next experiments while the first is described.

The paper mentions that the small amount of data is one of the problems. The authors may be interested in an older NoDaLiDa paper that presents a larger evaluation set [2].

Minor comments:

- Opening brackets can be made in LaTex using ` or ``
- Footnotes should immediately follow punctuation (not precede it)
- L434 that provides => which is (though this needs to be rewritten anyway, since the statement is not correct)
- L484 the AUC score


[1] Manjavacas, Enrique, and Lauren Fonteyn. "Macberth: Development and evaluation of a historically pre-trained language model for english (1450-1950)." Proceedings of the Workshop on Natural Language Processing for Digital Humanities. 2021.
[2] van Aggelen, A., Fokkens, A., Hollink, L. and van Ossenbruggen, J., 2019. A larger-scale evaluation resource of terms and their shift direction for diachronic lexical semantics. In Proceedings of the 22nd Nordic Conference on Computational Linguistics (pp. 44-54).

**Paper Type:**

Long paper

---

### Decision · Program_Chairs · 2023-03-17

Accept